# The identification of blood-derived response eQTLs reveals complex effects of regulatory variants on inflammatory and infectious disease risk

Claire Liefferinckx[1,2]*, David Stern[3], Hélène Perée[4], Jérémie Bottieau[1],
Alice Mayer[3], Christophe Dubussy[4], Eric Quertinmont[2], Vjola Tafciu[2],
Charlotte Minsart[1,2], Vyacheslav Petrov[4], Alex Kvasz[5], Wouter Coppieters[6], Latifa Karim[6],
Souad Rahmouni[4], Michel Georges[4,7], Denis Franchimont[1,2]

**1** Center for the study of IBD, Laboratory of Experimental Gastroenterology, Université libre de Bruxelles, Brussels, Belgium, **2** Department of Gastroenterology, Hepatopancreatology, and Digestive Oncology, HUB Hôpital Erasme, Université Libre de Bruxelles, Brussels, Belgium, **3** GIGA Bioinformatics Platform, GIGA Institute, University of Liège, Liège, Belgium, **4** Unit of Animal Genomics, GIGA Institute, University of Liège, Liège, Belgium, **5** Software development, University of Liège, Liège, Belgium, **6** GIGA Genomics Platform, GIGA Institute, University of Liège, Liège, Belgium, **7** WEL Research Institute & Faculty of Veterinary Medicine, Liège, Belgium

* claire.liefferinckx@hubruxelles.be

## Abstract

Hundreds of risk loci for immune mediated inflammatory and infectious diseases have been identified by genome-wide association studies (GWAS). Yet, what causal variants and genes in risk loci underpin the observed associations remains poorly understood for most. The identification of colocalized cis-expression Quantitative Trait Loci (cis-eQTLs) is a promising way to identify candidate causative genes. The catalogue of cis-eQTLs of the immune system is likely incomplete as many cis-eQTLs may be context-specific. We built a large cohort of 406 healthy individuals and expanded the immune cis-regulome through their whole blood transcriptome obtained after stimulation with specific toll-like receptor (TLR) agonists and T-cell receptor (TCR) antagonist. We report three mechanisms that may explain why an eQTL could only be revealed after immune stimulation. More than half of the cis-eQTLs detected in this study would have been overlooked without specific immune stimulations. We then mined this new catalogue of response (r) eQTLs, with public GWAS summary statistics of three diseases through a colocalization approach: inflammatory bowel diseases, rheumatoid arthritis and COVID-19 disease. We identified reQTL-specific colocalizations for risk loci for which no matching eQTL were reported before, revealing interesting new candidate causal genes.

## Author summary

Although many risk loci have been identified by GWAS for immune and infectious diseases, the causal variants and genes that underpin the observed associations remain unknown for most. The identification of colocalized cis-expression

**Data availability statement:** Data can be found at https://ega-archive.org/datasets/EGAD50000001320 Additionally, we provide an interactive web browser to facilitate user-friendly exploration of our dataset at https://tools.giga.uliege.be/cedar/publiclw.

**Funding:** CL received supporting grant from the Belgian Inflammatory Bowel Disease Research and Development group (https://www.birdgroup.be/en) CL received salary from Fonds National de la Recherche scientifique (https://www.frs-fnrs.be/en/) CL received salary and supporting grant from Fonds Erasme (https://fondserasme.org/fr/chercheurs) DF received supporting grants from Fonds National de la Recherche scientifique (https://www.frs-fnrs.be/en/) The funders had no role in study design, data collection and analysis, decision to publish, or preparation of the manuscript.

**Competing interests:** The authors have declared that no competing interests exist.

Quantitative Trait Loci (cis-eQTLs) is a recognized way to identify candidate causative genes. However, the catalogue of cis-eQTLs of the immune system is likely incomplete as many cis-eQTLs may be context-specific.

We built a large cohort of 406 healthy individuals and expanded the immune cis-regulome by analyzing the whole blood transcriptome after specific immune stimulations.

We observed that more than half of the cis-eQTLs revealed in our study would have been overlooked without specific immune stimulations. After characterization of this expanded cis-eQTL catalogue, we mined it with publicly GWAS summary statistics of three diseases using a colocalization approach. The specific exploration of reQTLs expanded the number of risk loci with matching eQTL and identified new candidate genes for inflammatory bowel diseases, rheumatoid arthritis and COVID-19. This work highlights the importance of using meticulously selected healthy cohorts at population based-level to improve our knowledge on the causal drivers of disease.

## Introduction

Genome-wide association studies (GWAS) have uncovered numerous risk loci for all studied common complex diseases [1]. Most of the underlying causative variants appear to be regulatory as opposed to coding, complicating the identification of the cognate causative genes [2,3]. Accordingly, identifying colocalized or matching cis-expression Quantitative Trait Loci (cis-eQTLs) in disease relevant tissues has proven an effective route to identify candidate genes in risk loci [4]. Cis-eQTLs are said to be matching if they exhibit SNP association patterns that are very similar to the SNP association patterns for the disease [5]. Previous studies have shown that matching cis-eQTL can be found for 25% of risk loci with the available eQTL data sets [6,7]. This raises the question as to the molecular drivers of the remaining risk loci. One possible explanation is that the matching cis-eQTL only manifest in specific, as of yet uninterrogated cell types and/or developmental stages. Another is that some of the matching cis-eQTL only manifest after onset of the disease process, such as in inflammatory conditions. Of note, most existing eQTL datasets were generated using heterogenous cohorts of healthy individuals.

To address the latter limitation, and in order to expand the catalogue of human cis-eQTLs that might underpin immune or infectious diseases, we studied the whole blood transcriptome of a highly selected cohort of 406 healthy individuals of white Europeans ancestry, both in resting state as well as after stimulation with specific toll-like receptor (TLR) agonists (TLR4 activated trough lipopolysaccharide (LPS) and TLR7/8 activated trough resiquimod (R848)) and T-cell receptor (TCR) antagonist (using anti-CD3 and anti-CD28 antibodies). The underlying assumption was that the corresponding perturbations might mimic inflammatory conditions and elicit novel

eQTLs (also known as "*response* eQTLs" or reQTLs) of which some may contribute to inter-individual variation in predisposition to immune mediated inflammatory (IMIDs) or infectious diseases [8–18].

## Results

### Leveraging both genome-wide association study and transcriptomics identifies genetic determinants of TCR stimulation

The GEOCODE cohort consists of 406 individuals, with details and methodologies outlined in Fig 1, including inclusion criteria and sample handling (see Methods). The inter-individual variability in circulating immune cell composition and baseline characteristics has been previously described [19]. We validated the effectiveness of the stimulations by assessing the levels of the corresponding cytokines (Table A in S1 Table). We applied an in-silico cell deconvolution technique [20] to assess the impact of immune stimulation on altering immune cell proportions. Interestingly, we observed that the proportions of immune cells remained unchanged after 24 hours of in vitro stimulation (S1 Data).

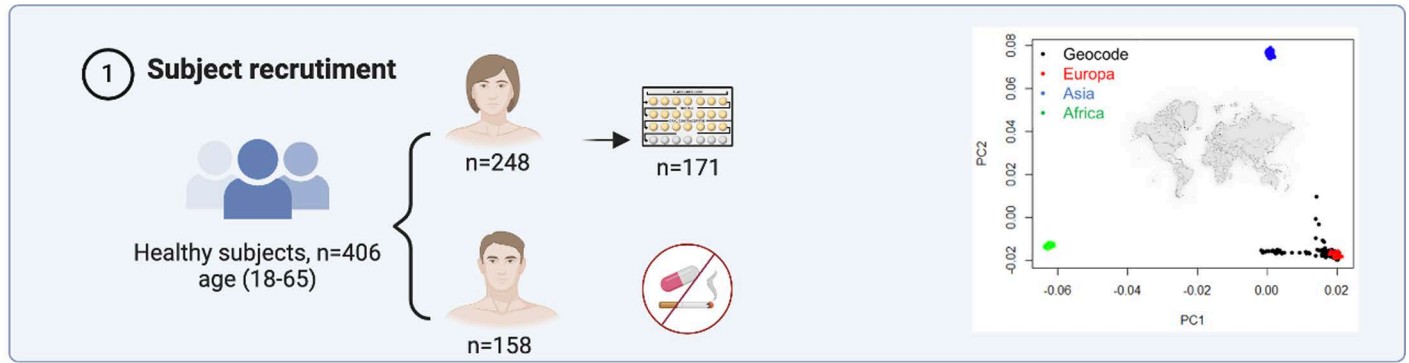

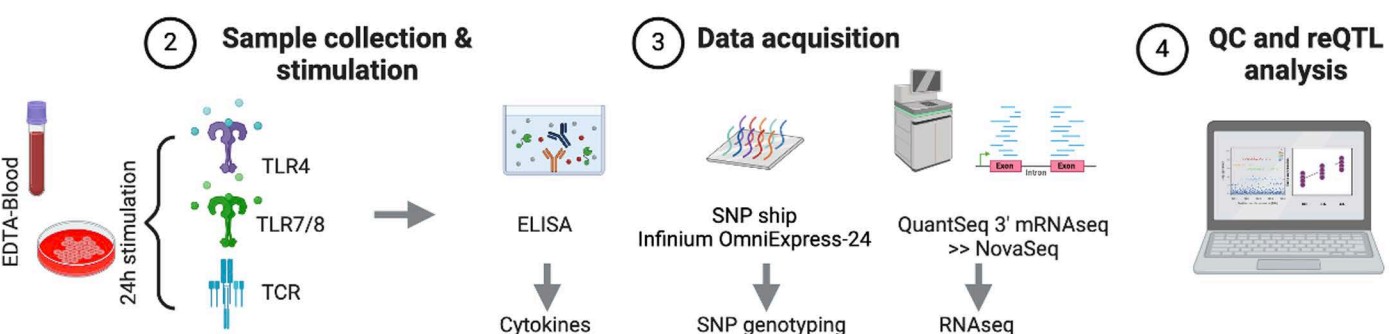

**Fig 1. GEOCODE study design. 1.Subject Recruitment**: The GEOCODE cohort consisted of highly selected healthy individuals with no history of disease, medication use, nor smoking habits. Among the 248 female participants recruited, 69% reported contraceptive pill use. The plot displays PCA mapping from the HapMap consortium for ancestry comparison alongside PCA mapping of the GEOCODE cohort. **2. Sample collection and stimulation:** Whole blood samples were collected, and cell cultures and stimulations were performed within three hours of collection. Three stimulation conditions were applied during a 24-hour incubation period: Resiquimod (TLR7/8 agonist), Lipopolysaccharide (LPS) (TLR4 agonist) Anti-CD3/anti-CD28 (TCR stimulation). **3. Data acquisition:** Cytokine production following TLR and TCR stimulation was quantified using a standard ELISA method, with all measurements expressed in pg/ml. Genotyping was performed using Illumina's Human OmniExpress BeadChips, and RNA sequencing was carried out with the QuantSeq 3' mRNA-Seq Library Prep Kit FWD at the GIGA-Genomics platform, resulting in a total of 1305 samples. **4. QC and reQTL analysis:** Cis-eQTL analysis was performed using QTLtools with 10,000 permutations to compute adjusted p-values. The resulting cis-eQTLs were processed following the guidelines outlined in reference 6. Images used in the figure have been generated with a Biorender Academic License.

All participants were genotyped with Illumina's OmniExpress SNP array and genotype data imputed to whole genome using TOPMed. Transcriptome analysis was conducted by RNA-Seq (Fig 1 and Methods). Gene expression profiles segregated samples in multidimensional expression space by conditions (Fig 2A-B). In general, principal component (PC) 1 separated stimulated samples (TLR7/8-TLR4-TCR) from controls, while PC2 separated TCR from TLR stimulated samples. However, we observed a continuum of individuals between TCR stimulation and control groups (Fig 2A). This is also evident in Fig 2B, where a subset of TCR stimulated samples (in purple) appears similar to the control group. Using a threshold correlation of |0.5|, PC1 was found to be influenced by 398 genes, with 61 showing positive and 337 showing negative correlation (Table B in S1 Table). Positively expressed genes highlighted Reactome pathways related to "Neutrophil degranulation" (R-HSA-6798695) and "Chemokine receptors bind chemokines" (R-HSA-380108) among the top significant pathways, while negatively expressed genes highlighted "Interleukin-10 signaling" (R-HSA-6783783) and "Inteferon alpha/beta signaling » (R-HSA-909733). The full list of pathways associated with these genes is presented in Table C in S1 Table.

The scale represents Euclidean distance, where shorter distances indicate higher correlations in gene expression between subjects. The Ctrl group (red) clearly segregates from the stimulated conditions. Among the stimulation conditions, the TLR-stimulated groups—LPS stimulation (TLR4, green) and R848 stimulation (TLR7/8, blue)—exhibit similar correlation patterns and are closer to each other compared to the TCR-stimulated group (anti-CD3/anti-CD28, purple), which forms a distinct cluster.

To further dissect the genetic architecture underlying this continuum, PCs were then calculated for each paired condition (control-stimulated condition). By considering individual PC values as new phenotypes in GWAS, rs1801274 on chromosome 1 was found to be significantly associated (p= 4e-45) with PC1 for TCR stimulation but not with other PCs (S1 Fig). As expected, 313 of the 398 genes driving PC1 obtained with the 4 groups were showing trans-eQTL effects for rs1801274 (Table B in S1 Table).

Rs1801274, is a missense mutation (G>A) (G wild type and A variant) in the FCGR2A gene on chromosome 1, encoding FcγRIIa receptor, a low-affinity receptor for the constant fragment (Fc) of immunoglobulin G (IgG) [21,22]. The A variant, common in Europeans (MAF G 0.49), results in an arginine (R) to histidine (H) substitution at position 131 or

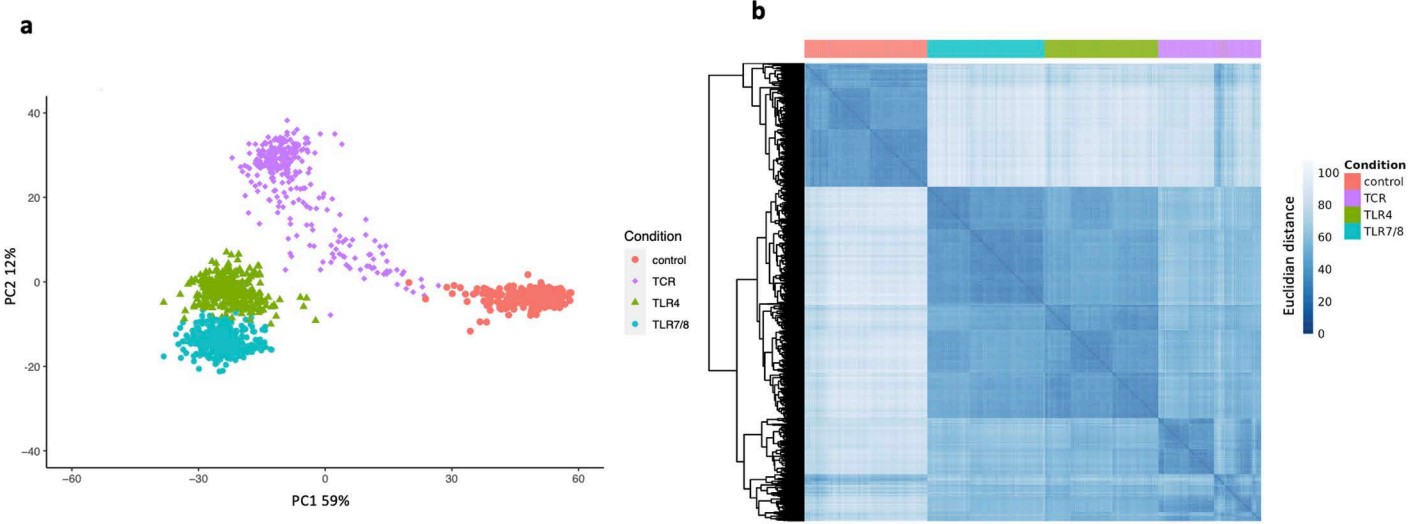

**Fig 2. Transcriptomic analysis. A. Principal component analysis (PCA)** of the expression of the most variable 500 genes across the four conditions of stimulation (Control (Ctrl), TLR4, TLR7/8 and TCR stimulations). **B. Heatmap depicting the correlation patterns of the 500 most variable genes across subjects after 24 hours of stimulation, as assessed by RNA-seq.**

166 (depending on the numbering in the literature) in the extracellular domain of the receptor protein, and was reported to impact the binding affinity for the Fc region of different IgG subclasses, affecting the cytokine production, clearance of immune complexes and phagocytosis of opsonized bacteria by granulocytes [23,24]. We confirmed that rs1801274 affect cytokine production and was strongly associated with IFNg (p=7.72e-40) and IL-2 levels (p=2.63e-15) following anti-CD3/anti-CD28 whole blood stimulation as reported by others [25,26] (S2 Fig). This suggests that part of the TCR stimulation in whole blood cell cultures might be mediated by the engagement of the FcγRIIA, expressed monocytes/macrophages, neutrophils and dendritic cells, by circulating IgG2 -containing immune complexes or released IgG immunoglobulins (including the co-stimulatory signal of the soluble anti-CD28 antibody). This underscores the role of monocytes/macrophages and granulocytes in the elicitation and perpetuation of the TCR stimulation thanks to the ex vivo model/assay of whole blood cell cultures. Interestingly, whole blood cell cultures specifically depleted in neutrophils demonstrated a reduction in IFNγ release, indicating that not only monocytes/macrophages but also neutrophils play a critical role in the cytokine release cascade through Fc:FcγR interactions [27].

### Expanding the immune cis-eQTL catalogue and clustering cis-eQTLs into cis-regulatory modules

We performed cis-eQTL analyses using QTL-tools, and gene expression levels pre-corrected for defined (sex, age, BMI) and hidden confounders (29–38 expression principal components). We identified a total of 13,679 autosomal cis-eQTL (FDR 0.05) corresponding to 6,496 eQTL genes (eGenes) (Tables D and E in S1 Table).

Cis-eQTL affecting the same gene in more than one condition were merged in a "cis regulatory module" (cRM) when sharing near-identical "association patterns", following ref. 6 (Figs 3A and S3). This yielded 8,401 cRM operating on average in 1.63 of the four tested conditions. cRM operating either in one specific condition or in all four conditions were the most common, accounting for 79.5% of the modules (Fig 3B). The predominance of modules operating in one condition only was unlikely to be due to limited cis-eQTL detection power, as allowing the merger of 12,950 non-significant cis-eQTL that would nevertheless match at least one significant cis-eQTL with 0.6 did not change the pattern (S4 Fig and Methods). It is noteworthy that 4,695 of cRM (55.8%) are not detected in resting condition, requiring either TLR or TCR stimulation to be detected.

We uncovered three mechanisms that may explain why a cRM may be active in condition "a" but not in condition "b". The first is that the target gene is expressed at too low levels in condition "b" for the eQTL to be detectable. Several genes exemplified this scenario such as IL36G and IL36RN associated with psoriasis [28], and CSF3 associated with severe neutropenia [29] (Figs 3C and S5). We estimate that this first scenario accounts for 2.6% of cases in our dataset (see Methods). The second mechanism is the loss of a genotype effect in condition "b", although the gene remains expressed at high enough levels for the eQTL to be detected if it exists. We estimated that this second scenario accounts for 72.8% of cases (see Methods). Examples of this scenario include NCF2 associated with chronic granulomatous diseases [30], and LILR2B (important immune mediator) [31] (Figs 3D and S6). Many eQTLs were specifically lost after "TCR" stimulation, and this was often associated with an increase in expression variance of the corresponding gene. One such example is the CISD1 gene associated with inflammatory bowel diseases (IBD) [32] (Fig 3D and S6). The third mechanism occurs when a gene (eGene) switches cRM between conditions: the regulatory variants governing gene expression in the four conditions, although possibly overlapping, are distinct enough to generate non-matching EAPs (Fig 3E). We estimated that this last scenario accounts for 16.5% of cases. Accordingly, 1,578 eGenes were assigned to two or more cRM (Table E in S1 Table). These comprised a high proportion of genes (55.7%) that were subject to one eQTL in resting condition, and another eQTL after stimulation, with occasionally distinct stimulation-specific EAPs (Figs 3E and S7). When a gene was influenced by the same cRM in more than one condition, the signs of the SNP effects were generally consistent across conditions (either up- or down-regulation). There were only few, yet noteworthy exceptions for which the allelic effects switched sign between control and TLR and/or TCR stimulation, including ADCY3, BMP8A and HIP1 (Fig 3F).

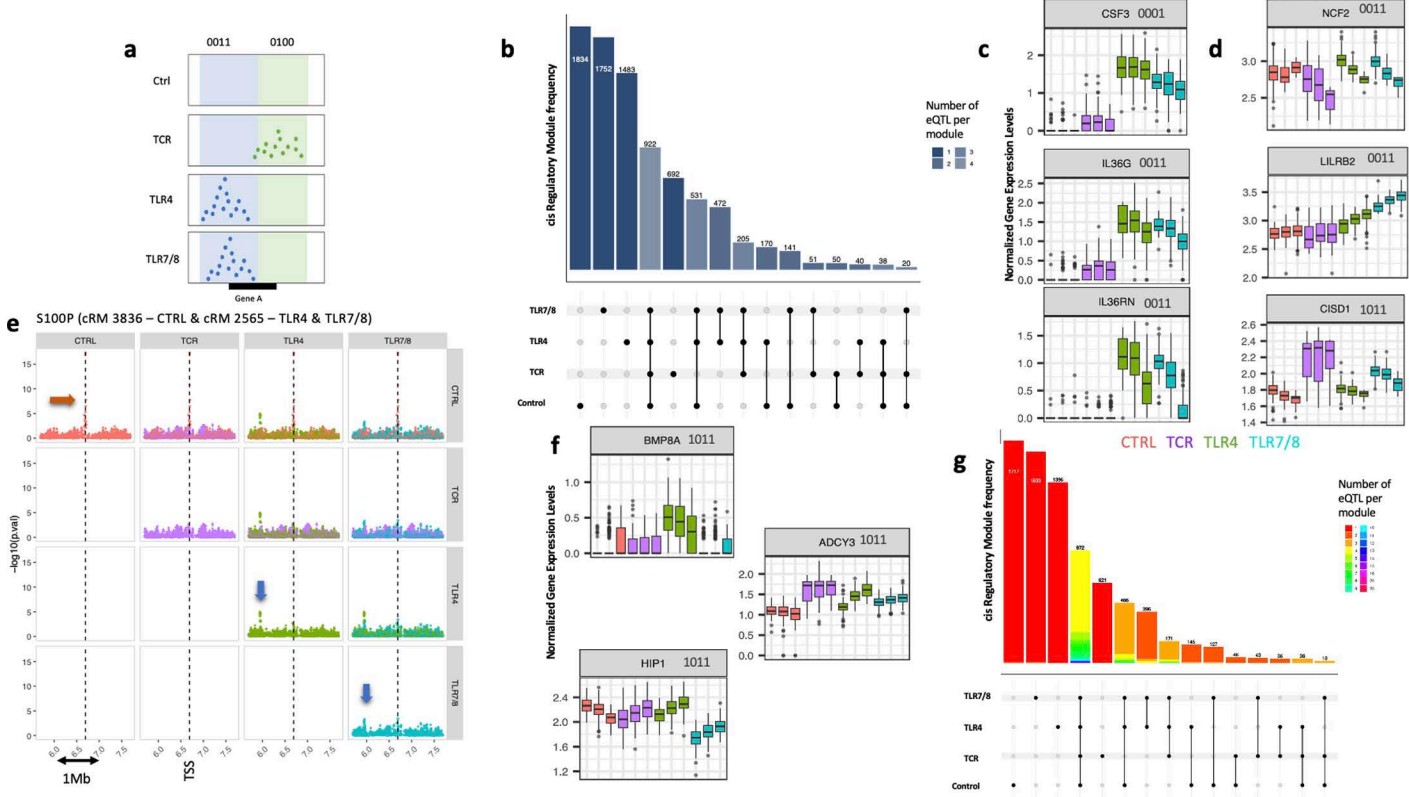

**Fig 3. Exploration of Response (R) cis-eQTLs in stimulated whole blood cell cultures from healthy subjects (GEOCODE cohort). A.** Schematic illustration of the eQTL colocalization methodology. The reQTL association patterns (EAPs) for TLR4 and TLR7/8 conditions are similar and are merged into the same cis regulatory module (cRM) associated with Gene A. This cRM is designated according to the following label 0011, indicating no eQTL in Control or TCR conditions, but identical reQTLs association patterns in TLR4 and TLR7/8 conditions. Gene A is associated with another eQTL whose EAP does not colocalized with the reQTLs for TLR4 and TLR7/8 conditions. This eQTL exists in its own cRM, labelled as 0100. **B.** Upset plot showing the distribution of Cis regulatory modules (cRMs) affecting the same gene under resting or TCR – TLR- conditions of stimulation. The plot highlights the predominance of condition-specific cRM with the majority being reQTLs. Since this graph only includes monogenic cRMs, the maximum number of eQTLs per gene is four, corresponding to the number of conditions in our dataset. **C.** First mechanism that may explain why an eQTL is active in one condition but not in an another condition: For each eQTL, the x-axis represents the four conditions with the three alleles per genetic variant. The expression level of the target gene's is too low in Ctrl (resting) condition for the eQTL to be detectable, but it becomes detectable under stimulation conditions. The Boxplots illustrate gene expression levels, with the y-axis showing normalized gene expression in arbitrary units and x-axis showing different stimulation (Control (Ctrl) in red, TCR in purple, TLR4 in green, TLR7/8 in blue). The genes CFS3 (Colony Stimulating Factor 3), IL-36G (Interleukin 36 Gamma) and IL-36RN (Interleukin 36 Receptor Antagonist) are used as examples. **D.** Second mechanism that may explain why an eQTL is only active in some conditions despite the expression of the gene being measured in all four conditions: For each eQTL, the x-axis represents the four conditions with the three alleles per genetic variant. This scenario occurs when a genotype effect is not consistently detected across all conditions, even though the gene is significantly expressed. For example, the cRM for the genes NCF2 (Neutrophil Cytosolic Factor 2) and LILRB2 (leukocyte immunoglobulin like receptor B2) is labelled 0011, indicating eQTLs are detected only in TLR7/8 and TLR4 conditions, although the gene expression levels are sufficiently high in Ctrl and TCR conditions for potential eQTL detection. Likewise, the cRM for the gene CISD1 (CDGSH Iron Sulfur Domain 1) is labelled 1011, indicating eQTLs are detected in Ctrl, TLR4 and TLR7/8 conditions but not in TCR condition. **E.** Third mechanism that may explain why an eQTL is active in one condition but not in another condition: The target gene switches between different cRM under various conditions. The plots illustrate the EAPs for the target gene S100P (S100 Calcium Binding Protein P) across different stimulation conditions, color-coded as follow: red for Ctrl, purple for TCR, green for TLR4, and blue for TLR7/8 conditions. The y-axis shows the distribution of −log(p) values for variants in the region around the top cis-eQTL, while the x- axis represents the genomic region centred on the Transcriptional Start Site (TSS) of the gene. The eGene S100P is influenced by two different cRMs. The peak indicating the first cRM, 3886, is highlighted by the red arrow, and the peak indicating the the second cRM, 2565, is highlighted by the blue arrow. In the latter case, the EAPs for the TLR4 and TLR7/8 conditions are similar and combined within a single cRM. **F.** Examples of cis-eQTLs where the direction of the allelic effect changes between the control and the conditions of stimulation. For each eQTL, the x-axis represents the four conditions with the three alleles per genetic variant. For the *BMP8A* gene (Bone Morphogenetic Protein 8a), the cRM is labelled 1011, indicating eQTLs detected in Ctrl, TLR4 and TLR7/8 conditions but not in the TCR condition. The eQTL effect

is positive in Ctrl and TLR7/8 conditions but negative in the TLR4 condition. For the *ADCY3 (Adenylate Cyclase 3) and HIP1 (Huntingtin Interacting Protein 1)* genes, the cRM is also labelled 1011. The eQTL effect is negative in Ctrl but positive in TLR7/8 and TLR4 conditions. G. Upset plot showing the distribution of Cis regulatory modules (cRMs) across resting, TCR and TLR conditions of stimulation, considering both mono- and multi- genic cRMs. The plot highlights the predominance of condition-specific cRM with most of them being reQTLs. Since the graph includes both monogenic and multigenic cRMs, the number of eQTLs per cRM can be as high as 25. A cRM can result from the merge of both several eQTLs linked to one gene and/or eQTLs linked to several genes. The red bars, representing single eQTL per cRM (monogenic and mono-condition eQTL), demonstrates that monogenic reQTLS are the most prevalent in our dataset.

We then merged the previously described "monogenic" cRMs that shared similar association patterns across genes (see Methods) (Fig 3G). This yielded 453 multigenic cRMs, encompassing an average of 2.38 genes per module. While modules that are only active following stimulation account for 56% (4296/7723) of all cRMs, they only account for 20% (92/453) of multigenic cRM (p < 10–6). Thus, stimulation-specific reQTL tend to be more gene-specific than eQTL that are also active under base-line conditions. Nevertheless, 36 multigenic cRM were activated by both TCR and TLR stimulation, and could have a large impact on the immune response (Table F in S1 Table). An open-access website has been developed to visualize cRMs within their genomic context (https://tools.giga.uliege.be/cedar/publiclw).

### Increasing the number of disease risk loci with matching cis-eQTLs

We then mined our catalogue for cis-eQTLs that colocalize with GWAS-identified risk loci for IMIDs and infectious diseases. We focused on inflammatory bowel diseases (IBD) [33] and rheumatoid arthritis (RA) [34], two IMIDs with worldwide growing prevalence, as well as on the infectious COVID-19 disease [35], a recent major healthcare issue, and involving the TLR7/8 signaling pathways [36]. Colocalization analyses were conducted following ref. 6. This analysis assumes that if a risk locus influences disease susceptibility by altering transcript levels of a given gene in cis, the disease association-pattern (DAP) for this locus and the EAP of the gene should be similar, quantified by the θ metric.

We found 187 significant DAP-EAP correlations (|θ| > 0.6, p 0.05) spanning 39 of the 244 IBD risk loci (16.2%) and encompassing 68 genes, as outlined in Table G in S1 Table. Of interest, 19.2% of these correlations (36 out of 187) were with reQTLs, including 11 risk loci that would have been overlooked in the absence of TLR and TCR stimulations. An intriguing example of the added value of immune stimulation is the long non-coding (lnc) gene *AJ009632.2* (with unknown function) which is subject to a cis-eQTL/ cRM active in both control conditions and after TLR4 stimulation. Yet, the effect on gene expression has opposite sign: the variants that increase IBD risk increase the expression levels of *AJ009632.2* in resting state, while doing the opposite after TLR4 stimulation (Fig 4A). The function of this lnc gene is for now poorly defined. *CTSS*, encoding cathepsin S, a lysosomal enzyme expressed by immune and epithelial cells in response to inflammation, regulating antigen presentation and taking part in degradation of extracellular matrix [37], is another interesting example. Its expression is modulated by one cRM in resting condition and a distinct cRM after TLR4 stimulation (S8 Fig). This *CTSS* TLR4 reQTL matched the DAP of the rs4845604 IBD risk locus, located on chromosome 1 (150,100,000–151,120,000). Additional examples of relevant DAP-EAP correlations linked to reQTLs that highlight new candidate genes worthy of further exploration in the context of IBD are listed in Table 1. In addition, even in resting conditions, the GEOCODE eQTL catalogue reveals new DAP-EAP correlations that shed light on previously unknown candidate genes (Table 2).

With regards to RA, we identified 56 significant correlations (|θ|>0.6, p 0.01) involving 14 of the 124 tested risk loci (11.3%) and 15 genes. Of interest, 12.5% of these correlations (7 out of 56) were associated with reQTLs, including 3 risk loci that would have been overlooked in the absence of TLR and TCR stimulations (Table H in S1 Table). Risk locus rs10497813, located on chromosome 2 (197,200,000–198,800,000), is an interesting region which has been associated with the risk to develop RA, where the DAP's top SNP corresponds to an intronic variant in the *PLCL1* gene [38]. Interestingly, the same region has been associated with the risk to develop Crohn's disease (rs6738825, Chr2: 197,310,000–198,110,000) [39]. DAP related to this risk locus matched reQTLs for *PLCL1* (Fig 4B), thereby confirming *PLCL1* as an interesting candidate gene to be further evaluated. This gene has recently been suggested to be linked to RA by modulating the inflammatory response in synoviocytes [40]. Strikingly, we observed that the sign of is positive for Crohn's disease,

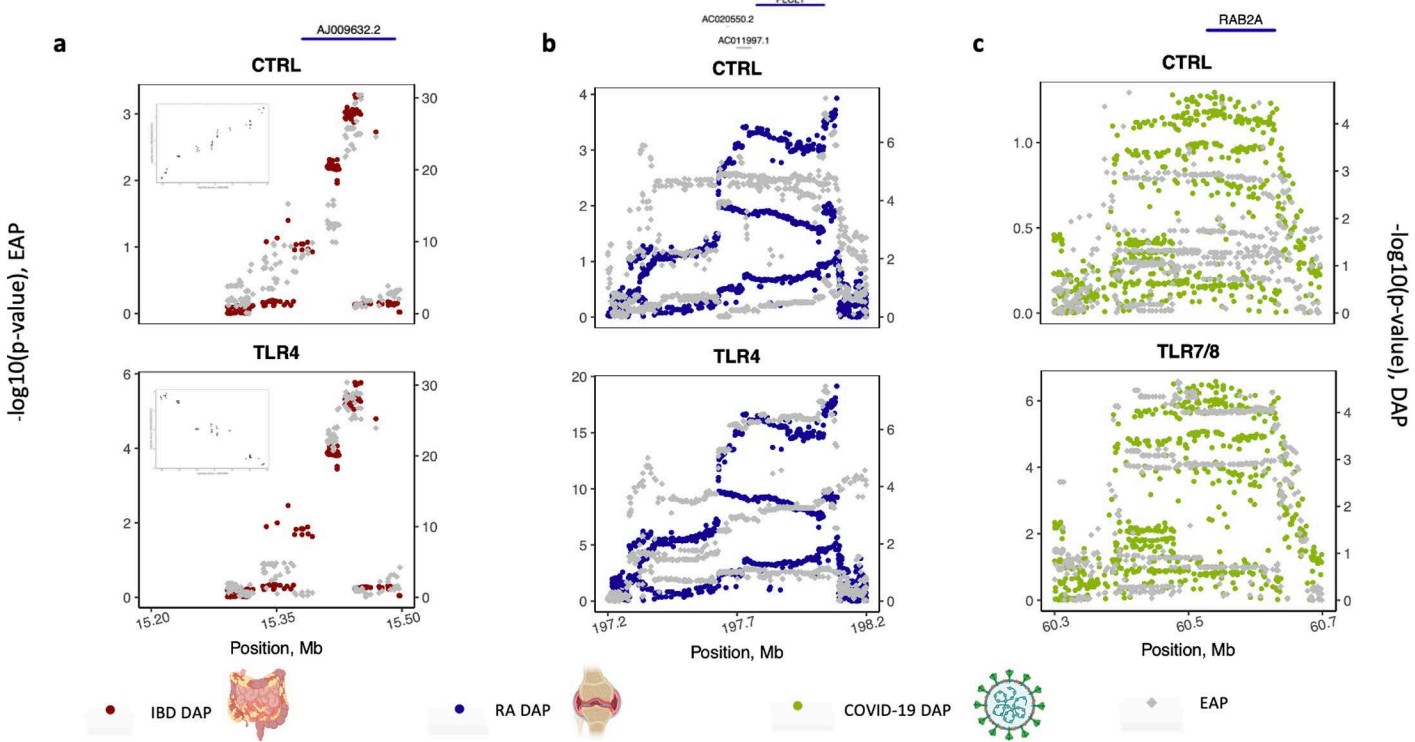

**Fig 4. Zoom plots of Disease Association Pattern (DAP) and eQTL association pattern (EAP). A. Zoom plot of a locus on Chromosome 21 displaying the correlation between a Disease Association Pattern (DAP) for Crohn's disease (Red) and an eQTL association pattern (EAP) for the AJ009632.2 gene (grey).** A p-value was calculated for the association between each surrounding SNP and the expression level of the *AJ009632.2* gene under different conditions of stimulation within a defined window around the target gene. The EAP on the **left Y-axis** represents the distribution of association −log(p) values for all variants within 1Mb of the eGene Transcriptional Start Site (TSS). Similarly, a p-value was extracted from GWAS summary statistics for each association between neighbouring variants of the top SNP and IBD phenotype. **The DAP** on the right Y axis represents the distribution of association −log(p) values for all variants within approximately 200 kb of rs2823286, an IBD risk locus from a public dataset. In this example, the DAP significantly correlates with the EAP for the Ctrl condition (θ = 0.91, p= 0.0004) and the EAP for the TLR4 condition (θ = -0.9, p= 0.001). The plots in the upper left panel show the correlation of each SNP in the EAP and DAP: the correlation is positive for the DAP-EAP in the Ctrl condition and negative for the DAP-EAP in the TLR4 condition. **B. Zoom plot of a locus on Chromosome 2 showing the correlation between a DAP for Rheumatoid arthritis (blue) and an EAP for PLCL1 gene (grey).** A p-value was calculated for the association between each surrounding SNP and the expression level of the PLCL1 gene within a defined window around this gene. **The EAP on left Y-axis represents** the distribution of association −log(p) values for all variants within 1Mb of the eGene Transcriptional Start Site (TSS), based on our dataset. Similarly, a p-value was extracted from GWAS summary statistics for each association between neighbouring variants of the top SNP and RA phenotype. The **DAP** on right Y-axis represents the distribution of association −log(p) values for all variants within approximately 200 kb of rs10497813, a RA risk locus from a public dataset. In this example, the DAP was not significantly correlated with the EAP for the Ctrl condition (θ = -0.04, p= NS) but was significantly correlated with the EAP for the TLR4 condition (θ = -0.73, p= 0.004). This example highlights that the eQTL linked to the *PLCL1* gene is a r*esponse* eQTL. **C. Zoom plot of a locus on Chromosome 8 illustrating the correlation between a DAP for COVID-19 (green) and an EAP for RAB2A gene (grey).** A p-value was calculated for each association between each surrounding SNP and the expression level of the RAB2A gene within a specifically defined window around this gene. The **EAP on the left Y-axis** represents the distribution of association −log(p) values for all variants within 1Mb of the eGene Transcriptional Start Site (TSS), based on our dataset. Similarly, a p-value was extracted from GWAS summary statistics for each association between neighbouring variants of the top SNP and COVID-19 phenotype. The **DAP** on right Y-axis represents the distribution of association −log(p) values for all the variants within the approximately 200 kb of rs2875968, a COVID-19 risk locus from a public dataset. In this example, the DAP was not significantly correlated with the EAP for the Ctrl condition but significantly correlated with the EAP for the TLR7/8 condition (θ = 0.73, p= 0.0001). This example highlights that the eQTL linked to *RAB2A* gene is a r*esponse* eQTL. Images used in the figure have been generated with a Biorender Academic License.

while it is negative for RA (Table 1 and S9 Fig). This suggests that increased expression of *PLCL1* in stimulated immune cells increases the risk to develop IBD, yet decreases risk to develop RA, and hence that the risk variants for IBD are protective for RA and vice versa. Additional examples of relevant DAP-EAP correlations that highlight new candidate genes worthy of further exploration in the context of RA are listed in Table 1.

**Table 1. Relevant reQTL-specific colocalisations with risk loci.**

| Locus | | Diseases | Gene | Best Colocalisations θ (p-val) | | | | Response eQTL | Ref. |
|---|---|---|---|---|---|---|---|---|---|
| Chr. | Coordinates | | | CTRL | TCR | TLR4 | TLR7/8 | | |
| Chr01_c | Chr1: 150,100,000–151,120,000 (rs4845604-a) | IBD, CD, UC | HORMAD1 | -0.02 (0.1) | -0.58 (0.01) | -0.82 (0.0004) | -0.82 (0.0009) | 1# | [57] |
| | | | CTSS | -0.05 (0.1) | 0 (0.4) | -0.72 (0.004) | -0.45 (0.03) | 1* | [58] |
| Chr01_f | Chr1: 161,050,000–162,050,000 (rs1801274) | UC | FCGR2A | -(-) | 0.75 (< 0.0001) | -(-) | 0 (0.9) | 1 | [59] |
| Chr02_a | Chr2: 197,310,000–198,110,000 (rs6738825-b) | CD | PLCL1 | 0 (0.4) | - (-) | 0.75 (0.005) | 0.64 (0.02) | 1* | [60] |
| | Chr2: 197,200,000–198,800,000 (rs10497813) | RA1,RA2 | PLCL1 | -0.04 (NS) | -0.73(0.0005) | -0.73 (0.0004) | -0.73 (0.0009) | 1* | [61] |
| Chr03_b | Chr3: 52,920,000–53,150,000 (rs2581828) | CD | RFT1 | 0.82 (0.006) | - (-) | 0.84 (0.005) | 0.22 (0.1) | 1* | [57–59] |
| Chr03_c | Chr3: 141,300,000–141,700,000 (rs724016) | CD | ZBTB38 | 0.39 (0.03) | - (-) | 0.73 (0.0007) | 0.48 (0.02) | 1* | [57] |
| Chr05_a | Chr5: 10,600,000–10,850,000 (rs2930047) | IBD | DAP | 0 (0.1) | - (-) | -0.59 (0.008) | -0.63 (0.005) | 1 | [58,59,62] |
| | Chr5: 10,500,000–10,900,000 (rs2918392) | RA1 | DAP | 0 (NS) | 0 (NS) | -0.23 (NS) | -0.63 (0.004) | 1 | [61] |
| Chr05_b | Chr5: 390,000–810,000 (rs4957048) | UC | TPPP | 0.01 (0.1) | - (-) | 0.8 (0.0005) | 0.68 (0.006) | 1 | – |
| Chr8 | Chr. 8: 60300000–60700000 (rs2875968) | COVID (B2,C2) | CA8 | - (-) | - (-) | -0.49 (NA) | -0.66 (0.002) | 1 | – |
| Chr12_a | Chr12: 111,200,000–112,600,000 (rs3184504) | RA1,RA2 | PHETA1 | 0 (NS) | -0.76 (NS) | 0 (NS) | - (-) | 1 | – |
| Chr12 | Chr. 12: 112800000–113150000 (rs10850094) | COVID (A2,B2,C2) | OAS1 | 0.34 (1) | 0.85 (0.0002) | 0.72 (0.004) | 0.91 (<0.0001) | 1 | [63,64] |
| Specific case of eQTL for which the effect on gene expression is opposite according to the condition of stimulation | | | | | | | | | |
| Chr21 | Chr21:15,290,000–15,500,000 (rs2823286) | IBD, CD, UC | AJ009632.2 | 0.91 (0.0004) | 0 (0.4) | -0.9 (0.0009) | -0.66 (0.01) | 0 | – |

The EAP-DAP matching listed in the Table 1 have been extracted from Tables F-H in S1 Table based on several criteria: 1. Only the EAP-DAP matchings with eQTLs defined as *Response* eQTL in our eQTL catalogue, 2. Only the EAP-DAP matching for which the DAP from risk loci showed strong p-values, 3. Only the EAP-DAP matchings for which the eQTL linked to the candidate gene was not extensively reported in the literature.

COVID-19 serves as another example, where we explored its pathogenesis by leveraging reQTLs generated following the activation of TLR7/8, a pathway relevant to the disease. Twenty-seven DAP-EAP correlations ($|θ|>0.6$ p 0.01) were observed, involving 5 risk loci (Tables 1 and I in S1 Table). Of note, 12 DAP-EAP comparisons, involving 2 risk loci, were captured through reQTLs alone and would have been overlooked without immune stimulation. The risk locus rs2875968 spans three genes (*RAB2A, LINC01301*, and *CA8*), none of which have strong prior evidence implicating them in COVID-19 physiopathology. Through our analysis, we identified eQTL associations between rs2875968 and the genes *RAB2A* and *CA8*, located on chromosome 8 (60300000 – 60700000) (Fig 4C). These eQTLs are active after TCR and TLR stimulation with opposite sign of effect on *RAB2A* and *CA8*. A recent study demonstrated that increased expression of *RAB2A* was linked to more severe COVID-19 outcome due to its role in viral replication [41]. In contrast, no current evidence supports an association between *CA8* and COVID-19 physiopathology. Beyond this example, colocalization analyses further validated the involvement of *IFNAR2* and *OAS1–3* genes in COVID-19 physiopathology (S10 Fig). Notably, the eQTLs related to the *OAS1–3* genes were part of a multigenic cRM active after both TCR and TLR stimulations.

PLOS Genetics

## Discussion

We built a large cohort of healthy individuals, stimulating their whole blood with TLR agonists and a TCR antagonist in a standardized approach to enhance our catalog of immune cis-eQTLs. We discovered that 72.9% of eQTLs, referred to as response eQTLs, were revealed through in vitro stimulation. We distinguished three potential mechanisms to explain the context-specific activity of some eQTLs. By exploiting our enhanced eQTL catalog to the analysis of risk loci for immune-mediated and infectious diseases, we demonstrated that response eQTLs improve our ability to identify new colocalizations and pinpoint candidate genes linked to these conditions.

Enhancing our eQTL catalog through functional studies offers a critical resource for mapping genetic effects on specific phenotypes. In such studies, carefully selecting participants is crucial to avoid noise introduced by disease, medications, lifestyle factors such as tobacco and alcohol use, and other independent health issues [42]. To date, only two large healthy cohorts have been established to investigate the genetic determinants related to stimulation-dependent immune response [43,44]. Our research has meticulously selected a highly homogeneous group of healthy participants—non-smokers, non-obese, under 65 years of age, not using any medications except for contraceptive pills, and with no history of chronic diseases or infections. This strategy minimizes external influences on the immune response. Furthermore, to reduce immune responses influenced by ex-vivo manipulations, without interfering with the cell-cell interactions that resides in their internal milieu, we used whole blood as an in-vitro model that best grasps the phenotype of the profile and magnitude of immune response of each individual [45]. This approach captures the overall immune response with minimal cell manipulation, although it may impair the detection of cell-specific eQTLs.

The majority of eQTLs captured in our study were response eQTLs, aligning with findings from other studies [9,11,13]. Notably, 72.9% of eQTLs were not detected in resting conditions, supporting the hypothesis that reQTLs are under significant selective pressure as supported by Kim-Hellmuth et al [13]. Additionally, response eQTLs were predominantly found in monogenic cis-regulatory modules, emphasizing their gene-specific behavior. Our findings also revealed that 1,578 eGenes were influenced by multiple independent eQTLs, with some genes regulated by distinct eQTLs under resting conditions and different eQTLs upon stimulation. This highlights how external stimuli can impact gene regulation through distinct genetic regulators.

As we identified three mechanisms underlying condition-specific cRMs, mechanistic hypotheses may help elucidate how immune stimulation triggers these genetic effects. Previous studies have shown that eQTLs frequently overlap with active cis-regulatory elements, such as enhancers and promoters, which interact with target gene promoters across various immune cell types [46]. Epigenetic modifications, including H3K27ac at enhancers and H3K4me3 at promoters, could drive chromatin accessibility changes, thereby influencing gene expression in a condition-specific manner. For example, LPS exposure has been shown to induce immune tolerance in monocytes through epigenetic modifications detectable as early as 1 hour post-stimulation [47]. Similarly, recent studies on chromatin accessibility in lymphoblastoid cell lines revealed that chromatin accessibility quantitative trait loci (caQTLs) can also account for immune-mediated disease associations. These findings highlight the critical role of chromatin accessibility variations in shaping the regulatory landscape that modulates gene expression [48]. These findings suggest that chromatin accessibility and histone modification dynamics could underlie the condition-specific activation of cRMs, potentially altering the detection or directionality of eQTLs under different conditions

We employed the previously described θ-based approach [6] for colocalization analysis. This method compares vectors of log(1/p) values (referred to as "association patterns" or APs) in a specific chromosomal region for pairs of traits under investigation. These pairs could involve two eQTLs (EAP1 vs. EAP2) or an eQTL and a disease (EAP vs. DAP). Importantly, the p-value vectors do not need to originate from the same cohort, as long as both cohorts represent the same population and thus share a similar LD structure. The comparison metric is Pearson's correlation coefficient, yet, (i) restricting analysis to SNPs with an association p-value < 0.05 for at least one trait, (ii) assigning greater weight to SNPs with stronger associations ("peaks" in the AP), (iii) incorporating the coherence of allelic effect signs across SNPs within traits,

and (iv) providing a signed θ value (positive or negative) to indicate, for example, whether increased gene expression correlates with increased (positive θ) or decreased (negative θ) disease risk. The statistical significance of θ is assessed by comparing the observed θ value with those generated through permutation (randomizing phenotype-genotype pairs) for one of the traits. Permutations are performed for both traits, and the final p-value is the average of the two empirical p-values. This approach accounts for variable LD patterns across the genome.

The -based approach is very similar to the SMR approach [49], with which it has been compared, and shown to provide comparable results [6]. In essence, the SMR approach performs a correlation analysis between the effect sizes of the SNPs () rather than their log(1/p)-values. Neither method aims to fine-map association patterns or resolve whether signals arise from one or multiple causative variants (allelic heterogeneity). For both approaches, concluding that two APs match, requires the same set of causative variants to drive both signals. This set of causative variants, of unknown size, is referred to as a regulatory module (RM). Importantly, two APs influenced by distinct but overlapping sets of regulatory variants will not be deemed as "matching." In this regard, both methods differ from other colocalization methods, including coloc [50–52] and eCAVIAR [53]. For example, coloc can exploit fine-mapping results (obtained f.i. using SuSiE [54]) and test for colocalization between pairs of multiple independent signals found to affect traits 1 and 2, respectively.

Colocalizing cis-eQTLs with genetic variants from the GWAS catalog has effectively pinpointed candidate genes within risk loci. We observed that many risk loci matching our eQTL dataset under resting condition were previously recognized, validating the relevance of our dataset (Tables G-I in S1 Table). Notably, even in resting condition, we discovered several significant eQTLs without prior links between the eGene and diseases as outlined in Table 2. Additionally, a substantial proportion of these new candidate genes (around 59%), corresponded to recently annotated genes, reflecting ongoing advancements in genome annotation and functional characterization. More importantly, by leveraging the power of our response eQTL dataset, we identified matches with risk loci that are typically missed without immune stimulation. This supports the hypothesis that some cis-eQTLs may manifest only under specific conditions.

Our study has several limitations. Firstly, as previously discussed, employing a whole-blood study design presents both advantages and constraints. Using whole blood for eQTL discovery offers us significant advantages, including accessibility, relevance for immune traits, and the ability to capture dynamic systemic responses [45]. However, we agree that whole blood also presents challenges, such as cell-type heterogeneity, potential signal dilution, and confounding factors that may reduce specific regulatory effects. To address these limitations, we performed in silico cell deconvolution techniques to refine the interpretation of our blood-based eQTL findings and to evaluate in which impact immune stimulation can alter immune cell proportions. Additionally, we verified that the detected eQTLs were only minimally influenced by blood cell traits. Secondly, our assessment of cis-eQTL effects was based on transcriptomic data collected after 24 hours of TLR and TCR stimulations. This timepoint, although beneficial for capturing a broad range of gene expressions, may overlook some genes activated in the initial phases of the immune response. Nonetheless, recent research reported that more genes were differentially expressed at 24 hours compared to 4 hours post-stimulation. Ultimately, we applied a colocalization method developed by our team [6], acknowledging that other existing colocalization methods might slightly alter the influence of the clustering of our eQTLs into cis regulatory modules as well as eQTL mapping with risk loci.

In conclusion, this work extended our understanding of the immune cis-regulome by conducting at a population-based level ex-vivo stimulations. After leveraging eQTL dataset in health, we applied a colocalization approach to public datasets of IMIDs and COVID-19 disease. The specific exploration of reQTLs expanded our ability to colocalize risk loci, which would have been overlooked under resting baseline conditions, thereby identifying new candidate for IBD, RA and COVID-19. The web-based browser that we developed here will facilitate the dissemination of our data with the research community.

**Table 2. Relevant eQTL-specific colocalisations with risk loci.**

| Locus | | | | Best Colocalisations θ (p-val) | | | | Response eQTL | Ref. |
|-------|---|---|---|---|---|---|---|---|---|
| Chr | Coordinates | Diseases | Gene | CTRL | TCR | TLR4 | TLR7/8 | | |
| Chr01_a | Chr1: 2,470,000–2,750,000 (rs10797432_a) | IBD,UC | TNFRSF14-AS1 | 0.79 (0.02) | 0.82 (0.008) | 0.85 (0.005) | 0.74 (0.02) | 0 | – |
| | | | AL139246.3 | -0.67 (0.05) | - (-) | -0.31 (0.07) | -0.76 (0.004) | 0 | – |
| Chr01_b | Chr1: 92,950,000–93,450,000 (rs34856868-b) | IBD | DR1 | -0.65 (0.0004) | -0.67 (0.0001) | -0.62 (0.001) | -0.71 (0.0004) | 0 | – |
| Chr01_c | Chr1: 150,100,000–151,120,000 (rs4845604-a) | IBD, CD, UC | CTSK | 0.15 (0.08) | -0.71 (0.006) | 0 (0.2) | 0 (0.3) | 0 | – |
| Chr02 | Chr2: 73,900,000–74,300,000 (rs28421442) | RA1 | AC073263.1 | -0.46 (NS) | 0.62 (0.005) | -0.69 (0.005) | -0.49 (NS) | 0 | – |
| Chr06 | Chr6:111,060,000–111,910,000 (rs3851228) | IBD, UC | Z97989.1 | -0.63 (0.004) | -0.46 (0.02) | -0.61 (0.005) | -0.67 (0.003) | 0 | – |
| | | | AL080317.2 | 0.75 (0.0005) | - (-) | 0 (NS) | - (-) | 0 | – |
| Chr06 | Chr6:166,860,000–167,230,000 (rs1819333) | CD | Z94721.3 | -0.5 (0.005) | -0.73 (0.0002) | -0.55 (0.002) | -0.46 (0.007) | 0 | – |
| Chr08_b | Chr8: 48,320,000–48,700,000 (rs7011507_b) | IBD,UC | AC026904.1 | 0.87 (0.0003) | 0 (0.6) | 0.72 (0.003) | 0.63 (0.007) | 0 | – |
| Chr11_b | Chr11: 64,050,000–64,500,000 (rs559928) | IBD, CD | AP003774.2 | -0.71 (0.0001) | -0.35 (0.02) | -0.66 (0.0003) | -0.71 (< 0.0001) | 0 | – |
| Chr13 | Chr13:40,800,000–41,000000 (rs7329174) | CD | AL590064.1 | -0.77 (0.02) | -0.68 (0.04) | -0.86 (0.004) | -0.61 (0.05) | 0 | – |
| Chr15 | Chr15: 76,800,000–77,600,000 (rs115284761) | RA1,RA2 | TSPAN3 | -0.78 (0.0007) | -0.66 (0.005) | -0.6 (NS) | -0.82 (0.0003) | 0 | – |
| Chr16_c | Chr16: 68,490,000–68,830,000 (rs1728785) | IBD, UC | AC126773.6 | 0.84 (0.006) | - (-) | 0 (0.4) | - (-) | 0 | – |
| Chr17 | Chr. 17: 39125000–40100000 (rs3744245) | COVID (B2) | ORMDL3 | -0.57 (NS) | -0.42 (NS) | -0.6 (<0.0001) | -0.6 (NS) | 0 | – |
| Chr21 | Chr21: 44,040,000–44,260,000 (rs7282490_b) | IBD,C-D,UC | AP001505.1 | 0.78 (0.004) | 0 (0.3) | 0 (0.2) | 0 (0.9) | 0 | – |
| | | | GATD3A | -0.79 (0.004) | -0.81 (0.002) | -0.66 (0.02) | -0.68 (0.01) | 0 | – |
| | | | TRAPPC10 | -0.83 (0.002) | -0.76 (0.005) | -0.8 (0.003) | -0.75 (0.006) | 0 | – |

The EAP-DAP matching listed in the Table 2 have been extracted from Tables F-H in S1 Table based on several criteria: 1 Only the EAP-DAP matchings for which the DAP from risk loci showed strong p-values,. 2. Only the EAP-DAP matching if there was no reported link between the genes and diseases on the basis of available eQTL information.

(i)The columns linked to "Locus" refer to the name and chromosomal coordinates of the corresponding risk locus (Locus, Chr, Beg, End) (GRCh38/hg19) as well as the disease where the locus has been reported (CD: Crohn disease, UC: Ulcerative colitis, IBD: Inflammatory bowel disease, RA: rheumatoid arthritis, COVID: COVID-19 disease). A risk locus was associated with a letter (ex: rs7282490_b) refers to a risk locus for which the window has been manually adapted before processing to EAP-DAP matching. If there is no letter associated with the Rs, that means that the boundaries of the risk locus have not been adapted.

(ii)The columns linked to "Best colocalizations" refer to the genes and conditions of stimulation involved in matching DAP–EAP ($|\theta| > 0.6$, p-val <0.05). Among the different EAP-DAP, the best θ-values and corresponding p-values are reported in the Table 2. (NS= non-significant p-value, (-) = no comput-ed DAP-EAP matching).

(iii)The column "Response eQTL" refers to the eQTL characterization as response eQTL (1) or not (0). Both the eQTL catalogue (Table C in S1 Table) and extended catalogue were interrogated to confirm the response eQTL status. Symbol "*" refers to eGenes with switched modules (including different cRMs: one cRM with resting conditions and one cRM with response eQTL), Symbol "#" refers to presence of eQTL in resting condition in the extended catalogue.

(iv)The column "Ref." results from an in-depth review of colocalization between immune -mediated inflammatory diseases (IMIDs) GWAS and reported immune eQTLs

## Materiels and methods

### Ethics statement

The study was approved by the Ethics committee of Erasme Hospital, Brussels, Belgium (Reference number: P2015/425, date approval: 03/11/2015). All used methods were in accordance with approved guidelines and were performed in accordance with the Declaration of Helsinki. Each subject signed an informed consent.

### Study population

The study cohort (GEOCODE cohort) involved 406 healthy subjects prospectively included between October 2016 and March 2018. Inclusion criteria included age between 18 and 65 years, smoking-free status, drugs free with exception of hormonal contraception or finasteride, benzodiazepine and proton-pump inhibitors, and self-assessment of being in "good health" (S1 Data). First, second-and third- degree family members were excluded as well as shift workers (chronic jet lag) or any of following situations within the previous two weeks of inclusion: active allergic disease, episode of body temperature > 38 °C, dentist consultation, endoscopy, vaccination or steroid (systemic or topic) treatment. Gender, age, height, weight, medical and familial history were collected for all participating subjects.

### Sample collection, whole blood cell culture, stimulations and cytokine measurements

For all participating subjects, 40 ml of peripheral blood (distributed in several tubes - see S1 Data) were collected between 07h30 and 10h00 a.m. (to standardize the circadian cycle) after overnight fasting. A fresh EDTA tube was processed within the same day of blood collection for immunophenotyping. Details on immunophenotyping are presented in S1 Data. All immune cell counts were expressed by mm3. Plasma and sera were aliquoted and stored at -20 °C for later measurements. Whole blood cell cultures and stimulations were performed within three hours of blood collection. Two different TLR agonists, Resiquimod (R848) - TLR7/8 and Lipopolysaccharide (LPS) - TLR4, and one TCR antagonist (anti-CD3/anti-CD28) were used (S1 Data). The IC50 dose (or concentration) of each stimulant was chosen from a ranging dose-stimulation pilot study. Briefly, the blood was diluted 1/4 with pre-warmed FBS-RPMI. TLR agonists or TCR antagonists were added prior to incubation for 24h, at 37°C in 5% $CO_2$ containing atmosphere. Whole blood from each well was next transferred into a pre-labelled 1.5 ml Eppendorf tubes. After centrifugation, the supernatants were collected and stored at -80°C until use. Cytokines production following TCR and TLR stimulation were measured using a standard ELISA method. IL-6 (DuoSet Human IL-6, R&D Systems) and TNFα (DuoSet Human TNF-α, R&D Systems) were measured for all TLR stimulation conditions. IFNγ (DuoSet Human IFNγ, R&D Systems) and IL-2 (DuoSet Human IL-2, R&D Systems) were measured for TCR stimulation conditions. All cytokine measurements were expressed in pg/ml.

### DNA extraction

Human genomic DNA was extracted from EDTA-collected peripheral blood by automated genomic DNA isolation Tecan Freedom. The extraction was performed in batches of 32 samples. Subsequently, concentration and quality DNA were measured by nanodrop ND-1000. DNA concentration standardized for all samples to 50 ng/ml.

### SNP genotyping and imputation

The 406 individuals were genotyped for >700 K SNPs using Illumina's Human OmniExpress BeadChips, an iScan system and the Genome Studio software (GIGA genomics core facility, Liège, Belgium). We eliminated variants with call rate ≤0.95, with the minor allele frequency (MAF) <0.05 and deviating from Hardy–Weinberg equilibrium (HWE) (<0.001). European ancestry of all individuals was analyzed by PCA using the HapMap population as reference and used the 3 first PC as covariates. We used The Michigan Imputation Server with TOPMed Imputation Reference panel (https://imputation.biodatacatalyst.nlhi.nih.gov) to impute genotypes at autosomal variants in our population. We removed all SNP with

info < 0.4. A new quality control has been applied after imputation by filtering all SNP with MAF <0.05 and HWE <0.001. Finally, 4,939,638 variants were obtained after imputation.

## GWAS Analysis

Before starting association analyses, several covariates were integrated for analyses, namely sex, age, Body Mass Index (BMI) and the three first principal components related to ancestry. The association analyses were based on linear regression. Wald test was used for phenotypes association. We corrected the P-value threshold for associations due to the multiple testing of phenotypes. Phenotypes and covariates were forced to follow a gaussian distribution, preserving only the original rank orders, by using a quantile normalization. After controlling for residual test statistic inflation via genomic control, Manhattan plots and quantile-quantile (QQ) plots were used to describe GWAS results.

## RNA extraction

RNA was extracted using Trizol Reagent (Thermo Fisher Scientific). Four conditions of stimulation at 24h were selected for each healthy individual (Control, R848, LPS, anti-CD3/antiCD28). In total, 1,574 samples were processed for RNA extraction. Briefly, Trizol Reagent obtaining phase was centrifuged at 3,300 g for 5 minutes and clarified phase was subsequently used to remove cell debris. Extraction was performed using RNeasy Plus Universal Mini Kit (QIAGEN). Samples were next loaded on QIAcube to finalize extraction. The quantification of RNA was performed using a Nanodrop spectrophotometer. The RNA quantity was expressed ng/µl with a median of 37 ng/µl (IQR 27.7 – 46.6). The quality of RNA was measured using QIAxcel. The median RNA Integrity Score was 7.2 (IQR 6.7 – 7.8). All samples that met criteria of having a RIS value of 6.0 or higher were batched for RNA sequencing.

## RNA sequencing

RNA sequencing was performed using QuantSeq 3' mRNA-Seq Library Prep Kit FWD at the GIGA-Genomics platform for a total of 1407 samples. Briefly, 100 ng of total RNA was used from each sample as the starting material. This method uses oligo dT beads to select poly-A mRNA from the total RNA sample. The selected RNA is then heat fragmented and randomly primed before cDNA synthesis from the RNA template and UMI was added before following steps. The resultant cDNA was next processed through Lexogen library preparation kit procedure following the instructions of the manufacturer and using designed indexed adapters for multiplexing of samples. After enrichment, the samples were qPCR quantified and equimolar pooled before proceeding sequencing on the NovaSeq equipment (Illumina), with a sequence coverage goal of 20M 150 bp single-end reads (median achieved was ~ 21M total reads). All samples were sequenced in batches with a mix of the four conditions (control and stimulated conditions) across the batches. A globin clear module was used to remove globin mRNA from whole blood conditions. The globin genes expression (HBA1, HBA2, HBB HBG1, HBG2, HBD) after using globin clear module was around 2% of total reads.

## RNA sequence QC and read filtering

Raw RNA-seq data were demultiplexed and trimmed with bcl2fastq then aligned with STAR using Homo_sapiens. GRCh38.97.gtf as reference genome. Reads sharing the same UMI and mapped to the same place in the genome were collapsed using UMI-tools to avoid bias related to qPCR amplification. Quantification of reads was evaluated by FeatureCounts. RNA-seq expression samples were scrutinized using several quality control measures before being included in the final analysis set. A variant calling was applied using QTLtools to evaluate correct sample matching. Samples which failed to QC and/or exposed matching discrepancies were removed for the final analyses. At the end of read filtering and QC, 1305 samples from 359 subjects were used for following analyses (354, 327, 334 and 290 in control, LPS, R848 and TCR groups, respectively).

From the 60,617 annotations included in the analysis, we excluded short RNAs, pseudogenes, and mitochondrial RNAs from the analysis, thus leaving 38,286 annotations. We next filtered out genes with zero CPM in 90% of all samples; leaving 22,272 annotations of which 15,495 were protein coding for downstream analysis.

## Transcriptomic analysis

The normalization of gene expression counts as well as the identification of differentially expressed genes were conducted using the DESeq2 v1.36.0 package, which accounts for library depth by calculating a normalization factor for each sample. The process involved the following steps: First, the geometric mean expression of each gene across all samples was computed to serve as a pseudo-reference. Second, the expression of each gene was divided by its pseudo-reference value to calculate ratios for each gene. The normalization factor for each sample was then determined as the median of these ratios specific to that sample. Finally, normalized counts were obtained by dividing the raw counts by the normalization factor. Differentially expressed genes were identified normalizing each pair of comparison (Stimulation vs Control condition) together.

Principal Component Analysis (PCA) was applied taking the 500 most variable genes across the samples to confirm stratification in the dataset regarding the group stimulations. Principal components (PCs) were analyzed to identify the genes with the greatest contribution to each PC, based on their loading values, which indicate the influence of each gene on the PC. Greatest contributor genes were selected accordingly to an absolute value of correlation coefficient to PCs higher than 0.5.

Pathway enrichment analysis was performed on the top differentially expressed genes in PC1 using the public Reactome database. Only biological pathways with a false discovery rate (FDR) <0.05 were included. A heatmap was used to visualize the correlation patterns of the 500 most variable genes across subjects after 24 hours of stimulation.

## Covariates

Previously to eQTL analysis, to remove confounding variation in the gene expression data, that might mask or skew the effects of local genetic variation, we calculated the expression principal components (PCs) for each of the four conditions. For each subset of eQTL analyses, the dominant expression PCs were chosen for inclusion as covariates in the model such that the number of significant associations was maximized across all conditions. Before inclusion of PC in analysis, we checked by GWAS if each PC was associated to a specific genetic signal. Gender, age, BMI and three top PC (from genotype data) were also considered as additional covariates. To gain insight into the biological meaning of these factors, the relationship between expression PC and phenotypic covariates were analyzed.

## eQTLs analysis and cis-regulatory modules (cRM)

The cis window was defined as 1 megabase up- and down-stream of the transcriptional start site (±1 Mb). Analysis was run by QTLtools using 10.000 permutations to get adjusted p-values. These adjusted p-values were then used to compute the corresponding false discovery rates (FDR or q-value). eQTLS are considered as true eQTLs regarding a FDR threshold ≤ 0.05. Of note, for the purpose of a specific analysis (see results), we also used an "extended" catalogue of eQTLs. The extended catalogue contains all eQTLs with an FDR lower or equal to 0.05 plus non significative eQTLs regarding the threshold of FDR but with a gene being part of a significative eQTLs in at least one of the four conditions.

For rs1801274, we conducted a trans-eQTL analysis using the same covariates as in the cis-eQTL analysis. The analysis was performed with QTLtools, employing 10,000 permutations to obtain adjusted p-values

The list of significant cis-eQTLs was used to delineate the corresponding eQTL association patterns (EAP). EAP was defined as the distribution of association −log(p) values for all the variants in the region of 1Mb around the eGene Transcriptional Start Site (TSS). Hence, the denomination of eQTL referred to the top significant variant of the corresponding EAP.

In a first step, we have analyzed all EAPs related to one given gene (monogenic) in a window of 1 Mb. If the expression of a given gene is influenced by eQTLs in two conditions, the corresponding EAPs are expected to be similar. In this case, the eQTLs are merged in a monogenic cis-regulatory modules (cRM). The maximum number of eQTLs merged in a cRM is equal to the number of conditions (that is 4, CTRL, TLR4, TLR7/8 and TCR conditions of stimulation). In a second step, we have analyzed all EAPs in a given window of 1 megabase. If the expression of different genes is influenced by the same eQTL, across different conditions, the corresponding EAPs are expected to be similar. In this case, the corresponding eQTL is a "multi-genes" eQTL. When "multi-genes" EAP are similar, the eQTLs are merged in multigenic cRMs. θ metrics was used to calculate the correlation between EAPs based on the methodology that we have developed and previously published [6], a minimal of 50 SNPs is considered for computing θ metrics. All EAP-EAP correlations with |θ metrics| ≥ 0.6 are considered for downstream analyses.

### Investigating the distribution of the three mechanisms driving cRM activity

Each monogenic cis-Regulatory Module (cRM) was characterized by indicating whether the gene is part of an eQTL (FDR ≤ 0.05) (1) or not (0) in the four conditions defined in order CTRL-TCR-LPS-R848. As an example, a gene matching as eQTL in the three conditions of stimulation but not in resting condition is labelled 0111 while another gene matching as eQTL in TCR and R848 conditions only is labelled 0101

We quantified the occurrence of changes in status for a gene regarding different conditions under specific scenarios.

First mechanism, a gene might be part of an eQTL in condition "a" but not in condition "b" due to a too low level expression in the second condition. We calculated the number of cases where an eQTL transitioned from 0 to 1, corresponding to the first mechanism, where gene expression levels are too low in a condition for the eQTL to be detectable. Next, we assessed cases where a gene match as an eQTLs in several conditions but is splitted in multiple modules, representing the third mechanism, where a gene (eGene) switches cRMs between conditions, the genes is part of an eQTL in both conditions but the involved variants are distinct. The remaining activity patterns were attributed to the second mechanism, the gene is detected in both conditions at sufficient levels of expression but only match as an eQTL in one condition. This second mechanism was validated using a permutation-based methodology approach fully described, including the computational and statistical procedures, here:https://doi.org/10.1101/2024.10.14.24315443.

### Correlations EAP and DAP

We used the disease association patterns (DAP) related to IBD, RA and Covid-19 risk loci reported in a recent GWAS meta-analyses [1–3]. If regulatory variants affect disease risk by altering gene expression, the corresponding DAP and EAP are expected to be similar, even if obtained from different cohorts with the same ethnicity. In the same approach than EAP-EAP correlations, all SNPs with nominal p-value <0.05 for at least one EAP, present in a specific window around the risk loci, were compared to EAP obtained in our dataset. For each DAP, we manually visualized the distribution of association −log(p) values to define the comparison window.

To assess whether some eQTL signals might be confounded by blood cell traits, we exploited data from recent GWAS meta-analyses of blood cell phenotypes [55]. Out of 7,122 identified loci, we extracted summary statistics for 2,162 loci associated with four key blood cell traits: eosinophils, neutrophils, lymphocytes, and monocytes. These statistics were used to generate plots illustrating the distribution of association −log(p) values for each locus. Given the large volume of plots, we focused on loci where the top SNP had an association exceeding −log(10^-15). Subsequently, we manually reviewed the distribution of association −log(p) values to define the comparison window, corresponding to a specific DAP. After validating the DAP window, we computed DAP-EAP in the same way than EAP-EAP previously described.

### Softwares

All analyses related to quality controls were performed using PLINK version 1.9 while PLINK version 2.0 was used for association analyses. Of note, PLINK version 2.0 can manage the imputation quality metric (described as phased

dosages meaning the probability of a true imputation) but not PLINK version 1.9. BCF tools v1.17 was used for filtering files obtained after imputation with INFO score. R software – package qqman was used for graphics. QTLtools 1.3.1 was used for eQTLs analysis following methodology developed in reference paper [56].

## Supporting information

**S1 Fig. Association of rs1801274 with genes related to PC1.** A. **Principal component analysis (PCA)** of the expression of the most variable 500 genes across the four conditions of stimulation (Control (Ctrl), TLR4, TLR7/8 and TCR stimulations). B. **Manhattan plot showing significant association of rs1801274 with PC1 phenotype.** Principal Components were calculated for expression of genes in each paired condition (control-stimulated condition). By considering PC as new phenotype in GWAS, rs1801274, located on chromosome 1, was significantly associated with PC1-TCR phenotype (p= 4e-45). C. **Distribution of rs1801274 genotypes in the TCR group**. TCR group observed in panel A has been highlighted by removing others groups and all individuals have been flagged by genotype of variant rs1801274. Individuals with genotype AA segregate from the two others genotypes (AG and GG) and tend to form the continuum with control group as shown panel A.
(TIFF)

**S2 Fig. Association of rs1801274 with IL-2 and IFNɣ levels following anti-CD3/anti-CD28 stimulation.** A. **Manhattan plot showing significant association of rs1801274 with IFNɣ level phenotype.** IFNɣ level was considered as phenotype and associated with genotypic data through GWAS. rs1801274, located on chromosome 1, was significantly associated with the phenotype (p= 7.72e-40). Geocode cohort available for this phenotype (n= 380). B. **Manhattan plot showing significant association of rs1801274 with IL-2 level phenotype.** IL-2 level was considered as phenotype and associated with genotypic data through GWAS. rs1801274, located on chromosome 1, was significantly associated with the phenotype (p= p=2.63e-15). Only a sub cohort was available for this phenotype (n= 187).
(TIFF)

**S3 Fig. Examples of monogenic and multigenic cis-Regulatory modules (cRMs).** A. **Examples of monogenic cRMs.** Graphical representation of one cRM modulating *UBE2L*3 (Ubiquitin Conjugating Enzyme E2 L3) and *C5* (Complement C5) genes. Every node corresponds to a condition-specific EAP linked to the Gene and edges connect pairs illustrates a significant matching of EAPs with θ ≥| 0.6|. B. **Examples of multigenic cRMs.** Graphical representation of two multigenic cRMs. The first module clusters eQTLs linked to *LLGL1* (LLGL Scribble Cell Polarity Complex Component 1) and *TOP3A* (DNA Topoisomerase III Alpha) genes. The second module clusters eQTLs linked to *TRGJP2* (T Cell Receptor Gamma Joining P2) and *TRGJP1* (T Cell Receptor Gamma Joining P1) genes. Every node corresponds to a condition-specific EAP linked to the genes and edges connect pairs illustrates a significant matching of EAPs with θ ≥ | 0.6|. The blue line is used in case of positive θ while red line is used in case of negative θ.
(TIFF)

**S4 Fig. Upset plot of the distribution of cis-eQTLs across resting and conditions of stimulation, based on an extended catalog including 26.629 cis-eQTLs extended.** Despite the larger catalog, the plot reveals the continued predominance of condition-specific eQTLs, with the majority being reQTLs. Since this graph only considers monogenic cRMs, the maximum number of eQTLs per gene is four, corresponding to the number of conditions in our dataset. The bars indicate the number of modules.
(TIFF)

**S5 Fig. Additional examples of the first mechanism.** These manually curated examples highlight eQTLs where the gene is either not expressed or expressed at too low level for the eQTL to be detected in the resting condition, but becomes detectable after stimulation. The boxplots display gene expression levels, with the Y-axis representing

normalized gene expression and the X-axis indicating different conditions (Control (Ctrl) condition in red, TCR stimulation in purple, TLR4 stimulation in green, TLR7/8 stimulation in blue).
(TIFF)

**S6 Fig. Additional examples of the second mechanism.** These manually curated examples highlight eQTLs where there is a loss of genotype effect in one specific condition (resting or one condition of stimulation), while the gene remains expressed at sufficiently high levels for the eQTL to be detected if it exists. X-axis shows the different conditions of stimulation (red – Crtl, purple – TCR, green – TLR4, blue- TLR7/8), and Y-axis shows normalized gene expression. The genes are indicated at the top of each panel. The cRM related to the target gene (in uppercase) are labelled 1000, 0100, 0010 and 0001 for the conditions of stimulation: Ctrl-, TCR-, TLR4 - and TLR7/8 -, respectively. The situations are depicted according to these labels.
(TIFF)

**S7 Fig. Manually curated examples of eQTLs when an eGene switches cRM between conditions.** The plots represent the EAPs for several selected genes under different conditions of stimulation (red – Crtl, purple – TCR, green – TLR4, blue- TLR7/8). The Y-axis shows the distribution of −log(p) values for all variants in the region around the top cis-eQTL. The X-axis represents a genomic region centered on the Transcriptional Start Site (TSS) of the gene. The selected eGenes are modulated by different cRMs. The peak of pattern of the first cRM is highlighted by a red arrow, while the pattern of the second cRM is highlighted by a blue arrow and when three different cRM exist, the pattern of the third cRM is highlighted by a green arrow.
(TIFF)

**S8 Fig. Specific example of the CTSS gene which is modulated by two distinct cRMs.** The dashed line represents the transcriptional start site (TSS) of the CTSS (Cathepsin S) gene. The Y-axis shows the distribution of −log(p) values for all the variants in the region around the top cis-eQTL. The X- axis represents a genomic region centered on the TSS of the CTSS gene. The EAP of the different conditions of stimulation are shown in colors (red – Crtl, purple – TCR, green – TLR4, blue- TLR7/8). The EAP in Ctrl condition (in red) is sufficiently different from EAPs in TLR4 and TLR7/8 conditions to be included in a specific cRM (cRM 4515). EAPs related to TLR4 and TLR7/8 conditions were similar and merged in a same cRM (cRM 2090).
(TIFF)

**S9 Fig. Zoom plots of Disease Association Pattern (DAP) and eQTL association pattern (EAP). A Zoom plot of a locus on Chromosome 2 illustrating the correlation between DAP for IBD (red) and EAP for PLCL1 gene (grey).** A p-value was calculated for each association between each neighbouring variants of SNP and the expression of the *AJ009632.2* gene within a specifically defined window around this target gene under each condition of stimulation. The **EAP,** shown on left Y-axis, represents the distribution of association −log(p) values for all the variants in the region in the 1Mb region around the eGene Transcriptional Start Site (TSS), derived from our dataset. Similarly, a p-value was extracted from GWAS summary statistics for each association between neighbouring variants of the top SNP and IBD phenotype. The resulting **DAP** on right Y-axis, represents the distribution of association −log(p) values for all variants in the ~200 kb region around rs6738825, an IBD risk locus from public dataset (ref 27). In this example, the DAP was not significantly correlated with the EAP linked to the Ctrl (θ = -0.04, p= NS) but was significantly correlated with the EAP linked to the TLR4 condition (θ = 0.75, p=0.005). **B Zoom plot of a locus on Chromosome 2 illustrating the correlation between DAP for RA (blue) and EAP for PLCL1 gene (grey).** A p-value was calculated for each association between each neighbouring variants of SNPs and the expression of the PLCL1 gene within a specific window around this gene. The **EAP on the left Y-axis,** represents the distribution of association −log(p) values for all variants in the 1Mb region around the eGene Transcriptional Start Site (TSS), derived from our dataset. Similarly, a p-value was extracted

from GWAS summary statistics for each association between neighbouring variants of the top SNP and RA phenotype. The resulting **DAP** on the right Y-axis, represents the distribution of association −log(p) values for all variants in in the ~200 kb region around rs10497813, a RA risk locus from a public dataset (ref. 25). In this example, the DAP was not significantly correlated with the EAP linked to the Ctrl condition ($\theta = -0.04$, p= NS) but was significantly correlated with the EAP linked to the TLR4 condition ($\theta = -0.73$, p= 0.004). The correlation plots show the correlation of each SNP in EAP and DAP. The add-on plots of correlation show that the correlation is positive for matching DAP-EAP in case of IBD but negative for matching DAP-EAP in case of RA. Images used in the figure have been generated with a Biorender Academic License.
(TIFF)

**S10 Fig. Zoom plot of a locus on Chromosome 12 showing correlation between DAP for COVID-19 (green) and EAP for OAS1 gene (grey).** For each neighbouring variant of the SNP, a p-value was computed indicating the association with the expression of the RAB2A gene within a specific genomic window. **The EAP, represented on left Y-axis,** was defined as the distribution of association −log(p) values for all variants within 1Mb vicinity around the eGene Transcriptional Start Site (TSS), derived from our dataset. Similarly, a p-value was extracted from GWAS summary statistics for each association between neighbouring variants of the top SNP and COVID-19 phenotype. The resulting **DAP** shown on right Y-axis, represents the distribution of association −log(p) values across all variants within a manually defined region ~200 kb around rs10850094, a COVID-19 risk locus obtained from public dataset (Ref. 26). In this particular example, the DAP exhibited no significant correlation with the EAP associated with Ctrl ($\theta = 0.34$, p= NS), whereas it displayed significant correlations with EAPs linked to TLR7/8 ($\theta = 0.91$, p <0.0001), TCR ($\theta = 0.85$, p=0.0002) and TLR4 conditions ($\theta = 0.72$, p=0.004)). This example underscores that the eQTLs associated with *OAS1* gene are r*esponsive* eQTLs. The add-on correlation plots demonstrate positive correlations between matching DAP-EAPs pairs. Images used in the figure have been generated with a Biorender Academic License.
(TIFF)

**S1 Table. DAP-EAP correlations about COVID-19 (summary statistics from Nature.** 2023;621(7977):E7-E26).
(XLSX)

**S1 Data. Exploring the deconvolution of whole blood tissue and influence on eQTL discovery.**
(DOCX)

## Acknowledgments

We gratefully thank all participants to the GEOCODE cohort.

## Author contributions

**Conceptualization:** Claire Liefferinckx, Denis Franchimont.

**Data curation:** Claire Liefferinckx, David Stern.

**Formal analysis:** Claire Liefferinckx, David Stern, Jérémie Bottieau, Christophe Dubussy.

**Funding acquisition:** Claire Liefferinckx, Denis Franchimont.

**Investigation:** Claire Liefferinckx.

**Methodology:** David Stern, Hélène Perée, Alice Mayer, Vyacheslav Petrov, Souad Rahmouni, Michel Georges.

**Project administration:** Claire Liefferinckx, Charlotte Minsart.

**Resources:** Eric Quertinmont, Vjola Tafciu, Charlotte Minsart, Wouter Coppieters, Latifa Karim.

**Software:** Alex Kvasz.

**Supervision:** Michel Georges, Denis Franchimont.

**Writing – original draft:** Claire Liefferinckx.

**Writing – review & editing:** Hélène Perée, Jérémie Bottieau, Charlotte Minsart, Souad Rahmouni, Denis Franchimont.

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
