## [Decision Letter · Decision Letter 0]

8 Nov 2024

PGENETICS-D-24-01090The identification of blood-derived Response eQTLs reveal complex effects of regulatory variants on Inflammatory and Infectious Disease riskPLOS Genetics Dear Dr. Liefferinckx, Thank you for submitting your manuscript to PLOS Genetics. After careful consideration, we feel that it has merit but does not fully meet PLOS Genetics's publication criteria as it currently stands. Therefore, we invite you to submit a revised version of the manuscript that addresses the points raised during the review process. Please submit your revised manuscript within 60 days Jan 07 2025 11:59PM. If you will need more time than this to complete your revisions, please reply to this message or contact the journal office at plosgenetics@plos.org. Please include the following items when submitting your revised manuscript:* A rebuttal letter that responds to each point raised by the editor and reviewer(s). You should upload this letter as a separate file labeled 'Response to Reviewers '. This file does not need to include responses to any formatting updates and technical items listed in the 'Journal Requirements' section below.* A marked-up copy of your manuscript that highlights changes made to the original version. You should upload this as a separate file labeled 'Revised Manuscript with Track Changes '.* An unmarked version of your revised paper without tracked changes. You should upload this as a separate file labeled 'Manuscript '. If you would like to make changes to your financial disclosure, competing interests statement, or data availability statement, please make these updates within the submission form at the time of resubmission. Guidelines for resubmitting your figure files are available below the reviewer comments at the end of this letter. We look forward to receiving your revised manuscript. Kind regards, Lin Chen, Ph.D.Academic EditorPLOS Genetics Anne O'Donnell-LuriaSection EditorPLOS Genetics

Aimée Dudley

Editor-in-Chief

PLOS Genetics

Anne Goriely

Editor-in-Chief

PLOS Genetics

**Journal Requirements:** **Additional Editor Comments (if provided):****Reviewers' comments:** Reviewer's Responses to Questions

**Comments to the Authors:**

Reviewer #1: Overview: Lieﬀerinckx et al. performed cis-eQTL mapping using whole blood samples from N=406 healthy participants across four conditions, including three different immune stimulation and baseline resting. Briefly, the authors discovered ~50% of eQTLs were exclusively found after stimulation treatment, i.e., response eQTLs, which highlighted that eQTL can be context-specific and potentially better explain disease risk. The authors performed colocalization analysis between conditions and between genes to identify cis-regulatory modules and characterize condition specificity of eQTLs. Finally, the authors performed colocalization with GWAS immune traits and identified novel genes that are missed by using resting samples alone. This work will be likely of interest to the statistical genetics and human inflammatory disease communities. This manuscript is written well, and results presented clearly. Overall, I suggest providing more details in describing methods. With that said I have several comments.

Major Comments:

1. For the results regarding association between rs1801274 and PC1-TCR, can the author provide more details in interpreting this finding? Given the PCs capture linear combinations of gene expression profile between individuals, can we interpret this as “trans-eQTL” effect in TCR condition? What’s interesting is that rs1801274 is missense variant for FCGR2A, but this gene was not found to be correlated with PC1-TCR (Supplemental Table 2). In fact, after running PCA on the gene expression matrix within each condition, the authors can look at the loading matrix output to determine which genes have highest loading (positive or negative) on the top PC, which reflect correlations.

2. It would be helpful for the author provides more explaination on how the eQTL merging method works. How is it different from colocalization analysis described in Wallace C. PLoS Genet. 2021? Does multiple cRM for a given gene means multiple causal eQTL for this gene?

3. In line with above, can the author provide interpretation on multigenic cRM? I am not surprised they are less common than monogenic cRMs given two genes have to be close enough to share the cis-eQTL association pattern. But it’s not clear how to interpret this in the context of reQTLs across conditions.

4. Regarding the three mechanisms to explain condition-specific cRM, can the author provide some mechanistic hypothesis? For example, could the treatment cause epigenetic markers changes such as h3k27ac for enhancers and h3k4me3 for promoters, or chromatin change (open to close or vice versa), such that detection of eQTLs can be specific to certain condition or the sign of eQTL can differ between conditions.

Minor Comments:

1. Figure 1A shows the separated clusters of four conditions, consistent with the expectation. To rule out the possibility that this may reflect batch effect, can the authors describe how RNA sequencing was done for these samples? Are they randomly assigned to different batches?

2. Are the axis for Supplemental Figure 1c correctly labeled?

3. For Figure 2B, please add units for values shown on the heatmap Figure 2B. Is it 0-100%? Please switch the color scale to 0 as white and 100 as dark blue, it’s very misleading right now. The color code doesn’t match the captions describing this figure. For example, control is in red (stated as green). Please make sure the color code is correctly described.

4. For Figure 3B, please choose 4 other different colors if you want to color it (not the same colors you used for four conditions), although I don’t think it’s necessary to color it as the upset plot is self-evident.

5. For Figure 3cdf, the x-axis looks like the allele count for risk allele, please annotate and describe the x-axis. Please label the units for y-axis.

6. Add label b in Figure 4.

7. The reference number in line 458 for EAP method looks like wrong. Should be (6) not (7).

Reviewer #2: Liefferinckx et al. presented their work on identifying cis-eQTLs that potentially regulate the expression of nearby genes in both resting and stimulated whole blood samples collected from a homogeneous group of healthy individuals. They identified response eQTLs specific to particular conditions and further provided three different scenarios to explain why a response eQTL was observed under stimulation. Lastly, they overlapped the eQTLs with publicly GWAS data on immune related diseases to identify candidate genes.

My main concerns are as follows:

The authors used whole blood samples rather than sorted specific immune cell populations, such as dendritic cells (reference 8) and monocytes (references 8 & 11) for the innate immune response or T cells (reference 9) for adaptive immune response, as in previous work. Given that differences in abundances of specific immune cells and other blood cell traits across individuals have a genetic basis (Vuckovic et al 2020: https://doi.org/10.1016/j.cell.2020.08.008), could the eQTL signals that were identified in whole blood be confounded by blood cell traits? How would it potentially affect the identification of response eQTLs? I understand that performing cell sorting is not feasible at this stage, but have the authors considered using gene expression signatures to infer cell type composition, and assess its effects on the identification of (r)eQTLs? In addition, the authors should discuss the specific advantages and constraints of using whole blood in lines 289-290.

Secondly, there is lack of methods for identifying response eQTLs. Are these defined simply as eQTLs that passed multiple testing correction in one condition but not in another, or are more formal statistical tests performed? The former is less ideal since an eQTL that showed suggestive evidence but did not pass multiple testing correction in other conditions (often due to limited power) could still be potentially shared across conditions, and thus not a true reQTL. Also, are directions of effects considered when defining reQTLs? For example, are eQTLs with opposite effect signs considered context specific eQTLs?

Thirdly, the authors proposed three scenarios to explain context specific eQTL signals. It would be helpful to estimate the proportion that can be attributed to each of the three scenarios in this dataset, and to provide some comments on whether it can be generalisable to other cohorts/dataset.

Lastly, it appears that more eQTLs were identified in the resting samples only compared to stimulated samples (more control specific eQTLs as shown in Figure 3b and 3g; please provide specific numbers if it's not the case). Is this unexpected to the authors too? Given that control samples showed a high proportion of genes with low expression levels (figure 2B), one would expect that it leads to fewer eQTLs as expected under the first mechanism proposed by the authors.

Minor comments:

Discuss what’s novel from this study compared to previous response eQTL studies.

Line 112: I don’t think I can find how the driven genes were identified in Methods. Are these driven genes enriched in relevant pathways?

Line 458: please confirm if the correct reference is 6 or 7.

Line 424: please provide more details on the processing of the gene expression data. For instance, how the data were normalised, was the normalisation performed within each condition or across conditions, how correction for covariates was performed (before or after normalisation), etc.

Some work is needed to improve the readability:

• Supplementary Figure 1: what does the x-axis represent in panel C? The y-axis indicates PC1 but panel C looks identical to the TCR group in panel A; however, I think the authors calculated PCs in each condition separately for this GWAS. Could the authors please clarify?

• Figure 2B showing expression heatmap: Could the authors explain what the scale from 0 to 100 means? LPS stimulation (TLR4) is green, not plink as stated in the current legend.

• Could the authors make text in Figure 3 (especially b and g) bigger?

• Figure 3E: the legend refers to a red arrow but I can’t see any

Regarding data sharing: would the summary statistics for cis-eQTLs, response eQTLs, the merged cis regulatory module (cRM) that authors defined in this study, and colocalised or matched eQTL – GWAS signals be made publicly available? These data should not involve any sensitive information, and they would be valuable for future research.

**Have all data underlying the figures and results presented in the manuscript been provided?**

Reviewer #1: **No: ** Raw data not available and made upon request only. Summary statistics data is in supplemental table

Reviewer #2: Yes

PLOS authors have the option to publish the peer review history of their article (what does this mean? ). If published, this will include your full peer review and any attached files.

**Do you want your identity to be public for this peer review?** For information about this choice, including consent withdrawal, please see our Privacy Policy .

Reviewer #1: No

Reviewer #2: No

---

## [Decision Letter · Decision Letter 1]

29 Jan 2025

Dear Dr Liefferinckx,

We are pleased to inform you that your manuscript entitled "The identification of blood-derived response eQTLs reveals complex effects of regulatory variants on inflammatory and infectious disease risk" has been editorially accepted for publication in PLOS Genetics. Congratulations!

Yours sincerely,

Lin Chen, Ph.D.

Academic Editor

PLOS Genetics

Hua Tang

Section Editor

PLOS Genetics

Aimée Dudley

Editor-in-Chief

PLOS Genetics

Anne Goriely

Editor-in-Chief

PLOS Genetics

Comments from the reviewers (if applicable):

Reviewer's Responses to Questions

**Comments to the Authors:**

Reviewer #1: The authors addressed all my comments.

Reviewer #2: The authors have significantly improved the manuscript.

My remaining comments regarding the responses to my first comment:

The authors showed that the estimated proportions of cell lines are quite similar across four conditions, which is reassuring (only in terms of the robustness of eQTL detection; not saying I definitely expected the results). That said, this does not entirely rule out the possibility that some eQTLs may be tagging the change in cell type composition. I think a more direct test would be accounting for the cell type proportions as a covariate in eQTL mapping and see if this affects the identification of eQTLs, although I doubt it would have a huge impact given what has been shown by the authors.

The fact that the authors found rs1801274 again in their GWAS of changes in cell type proportions suggests to me that PC1 might capture some of the variation in cell type composition and the association between rs1801274 and PC1-TCR could simply reflect this (relevant to the other reviewer’s first comment). I agree with the authors that we won’t have a definitive answer using data from the current study design.

**Have all data underlying the figures and results presented in the manuscript been provided?**

Reviewer #1: Yes

Reviewer #2: Yes

PLOS authors have the option to publish the peer review history of their article (what does this mean? ). If published, this will include your full peer review and any attached files.

**Do you want your identity to be public for this peer review?** For information about this choice, including consent withdrawal, please see our Privacy Policy .

Reviewer #1: No

Reviewer #2: No

**Data Deposition**

http://datadryad.org/submit?journalID=pgenetics&manu=PGENETICS-D-24-01090R1

**Press Queries**

---

## [Editor Report · Acceptance letter]

PGENETICS-D-24-01090R1

The identification of blood-derived response eQTLs reveals complex effects of regulatory variants on inflammatory and infectious disease risk

Dear Dr Liefferinckx,

We are pleased to inform you that your manuscript entitled "The identification of blood-derived response eQTLs reveals complex effects of regulatory variants on inflammatory and infectious disease risk" has been formally accepted for publication in PLOS Genetics! Your manuscript is now with our production department and you will be notified of the publication date in due course.

With kind regards,

Anita Estes

PLOS Genetics

On behalf of:
